# The Future of Chronic Kidney Disease Treatment: Combination Therapy (Polypill) or Biomarker-Guided Personalized Intervention?

**DOI:** 10.3390/biom15060809

**Published:** 2025-06-03

**Authors:** Sajjad Biglari, Harald Mischak, Joachim Beige, Agnieszka Latosinska, Justyna Siwy, Mirosław Banasik

**Affiliations:** 1Mosaiques Diagnostics GmbH, 30659 Hannover, Germany; biglari@mosaiques-diagnostics.com (S.B.); latosinska@mosaiques-diagnostics.com (A.L.); siwy@mosaiques-diagnostics.com (J.S.); 2Department of Nephrology, Transplantation Medicine and Internal Diseases Institute of Internal Diseases, Wroclaw Medical University, 50-551 Wroclaw, Poland; miroslaw.banasik@umw.edu.pl; 3Kuratorium for Dialysis and Transplantation (KfH) Leipzig, 04129 Leipzig, Germany; joachim.beige@kfh.de; 4Department of Nephrology, St. Georg Hospital Leipzig, 04129 Leipzig, Germany

**Keywords:** chronic kidney disease, therapeutic interventions, combination therapy, personalized medicine, biomarker-guided therapy, treatment response prediction

## Abstract

Chronic kidney disease (CKD) is a global health burden that affects close to one billion individuals. As many healthcare systems struggle to accommodate existing patients, CKD incidence and related costs are projected to continue rising. Based on a systematic search, this narrative review offers an in-depth assessment of advances in CKD pharmacotherapy published between 2020 and 2025, with a specific emphasis on drug combinations. Various treatment approaches for CKD exist, many of them targeting different mechanisms. Therefore, combining multiple medications could provide patients with better outcomes, though this comes with the risk of increased adverse effects and unnecessary costs. Alternatively, using biomarkers presents an opportunity to ascertain the most appropriate treatments specifically tailored to an individual’s molecular profile, thus personalizing CKD management. The second part of this review presents the current state-of-the-art methods to guide CKD therapy based on markers predicting treatment response. Collectively, this review presents possible pathways toward more effective CKD treatment.

## 1. Introduction

Chronic kidney disease (CKD) is a common and progressive condition that has emerged as a leading global health burden, estimated to affect over 800 million people [1]. Large meta-analyses report a CKD prevalence of around 10–13% (stages 1–5) and approximately 5–10% for moderate-to-severe CKD (stages 3–5) [1]. Prevalence varies with region and risk factors; CKD is more common in older individuals, women, and people with diabetes or hypertension [1]. Despite advances in care, CKD-related mortality continues to rise; global death rates from CKD increased by 41.5% between 1990 and 2017 [1]. CKD is now ranked among the top causes of death worldwide (12th in 2017) and is projected to become the 5th leading cause of years of life lost by 2040 [1]. Notably, these figures do not even account for deaths indirectly attributable to CKD (e.g., cardiovascular deaths, the leading cause of overall mortality worldwide, to which CKD strongly contributes) [1]. According to age- and sex-adjusted data, 33.3% of patients with mild to moderate CKD (eGFR 45–59 mL/min/1.73 m^2^) and up to 39.9% of patients with moderate to severe CKD (eGFR 15–29 mL/min/1.73 m^2^) died from CVD, compared to 26.0% of patients with eGFR ≥ 60 mL/min/1.73 m^2^ without proteinuria [2]. This growing multifaceted burden underscores the urgent need for improved prevention, early detection, and innovative, personalized treatment and management strategies for CKD [1], supported also by the recent call to action by the Stronger Kidneys Taskforce of the European Renal Association, the European Kidney Health Alliance, and the European Kidney Patients Federation [3].

The sheer number of CKD patients and the progressive nature of the disease present enormous individual and public health challenges. Despite a CKD prevalence of approximately 10% in Germany, awareness among affected individuals remains alarmingly low; CKD unawareness was 71% in stage 3a, 49% in stage 3b, and still 30% in stage 4, hindering timely diagnosis and treatment [4]. Patients frequently present at late stages or with kidney failure requiring replacement therapy (dialysis), which is both life-altering and costly (Figure 1). Ensuring early detection and providing guideline-directed care could substantially reduce CKD progression and complications [5,6].

In response to this escalating global burden, the 2024 KDIGO (Kidney Disease: Improving Global Outcomes) Clinical Practice Guideline for CKD expands on earlier recommendations and urges clinicians to adopt a more individualized, risk-based approach [7]. Specifically, it reaffirms the glomerular filtration rate (GFR) and albuminuria categories while prioritizing validated prediction models, such as the Kidney Failure Risk Equation (KFRE), to guide intervention intensity [7]. However, GFR and albuminuria are consequences and not causes of CKD and cannot be effective for implementing CKD prevention strategies. Health disparities also impede progress; limited access to nephrology care and essential medications in many regions contributes to high avoidable mortality. It is estimated that over a million deaths per year from reversible acute kidney injury (AKI) occur due to a lack of access to dialysis and related care [5].

Managing CKD and its complications imposes a heavy economic burden on healthcare systems, and annual healthcare costs rise exponentially in advanced cases. For example, one analysis across 31 countries found the mean annual cost per patient was about $3000 at CKD stage 3a but can escalate to $57,000 at stage 5 (hemodialysis) [8]. Kidney transplantation incurs high upfront costs (>$75,000 in the first year) but lower maintenance costs thereafter (around $17,000) [8]. The estimated total yearly costs in these 31 countries are likely US$ 400 billion [8]. CKD is responsible for the most significant number of people experiencing catastrophic healthcare expenses each year, especially in low-resource countries [5]. These costs stem not only from dialysis and transplantation but also from managing CKD’s frequent complications. For example, CKD dramatically increases the risk and expense of cardiovascular events; a single hospitalization for myocardial infarction or stroke in a CKD patient can cost between $10,000 and $20,000 [8].

CKD results from the interplay of multiple injury pathways that lead to irreversible nephron loss and fibrosis. Regardless of the initial etiology (diabetes, hypertension, glomerulonephritis, etc.), progressive CKD is characterized by maladaptive activation of the kidney’s hemodynamic, inflammatory, and fibrotic pathways [9]. In diabetic and hypertensive nephropathy, hyperfiltration (GFR > 125 mL/min/1.73 m^2^) causes glomerular injury via afferent arteriolar vasodilation, efferent vasoconstriction, and tubuloglomerular feedback suppression [10]. This is coupled with persistent activation of the Renin-Angiotensin-Aldosterone System (RAAS) and Angiotensin II secretion, which maintains elevated glomerular pressures and promotes sodium retention, hypertension, and further nephron damage, in turn accelerating fibrogenesis via increasing transforming growth factor-beta (TGF-β) release, inflammatory cell recruitment, and extracellular matrix accumulation, culminating in glomerulosclerosis and tubulointerstitial fibrosis [11]. However, clinical trials targeting TGF-β directly (e.g., monoclonal antibodies) have failed, likely due to its dual role in latent (protective) and active (pathogenic) forms [12].

Chronic inflammation is integral to CKD pathology, with injured cells releasing chemokines and danger signals that recruit immune cells, perpetuating a cycle of inflammation and fibrosis [11]. Elevated inflammatory cytokines, including interleukin-6 (IL-6) and tumor necrosis factor-alpha (TNF-α), correlate with disease severity and contribute to ongoing renal deterioration [11]. Hypoxia can induce an epigenetic “memory” that perpetuates fibrosis, whereas oxidative stress amplifies NF-κB-driven inflammation [13].

Fibrosis, the hallmark of CKD, results from excessive collagen deposition, impaired degradation [14], and the sustained activation of fibroblasts and myofibroblasts [15]. These alterations drive myofibroblast proliferation, macrophage infiltration, and excessive extracellular matrix deposition, leading to irreversible tissue scarring [15]. A self-perpetuating cycle is driven by fibroblast/pericyte differentiation, macrophage infiltration, and profibrotic mediators (e.g., TGF-β, WNT/β–catenin) from injured tubules [15] (Figure 2).

The recognition that CKD shares pathogenic links with cardiovascular and metabolic disease has led to a “cardiovascular–kidney–metabolic” framework for research [9], highlighting the need for interdisciplinary and multifactorial care. This underlines that managing CKD requires controlling blood pressure, metabolism, and systemic inflammation, ideally addressing individualized molecular targets. Due to these diverse pathophysiological mechanisms involved in CKD, monotherapies often fail to achieve optimal outcomes. As a result, combination therapy and biomarker-guided personalized medicine have gained attention as promising approaches for effective CKD management.

This review provides an up-to-date overview of the recent advances in CKD management (2020–2025). Established therapeutic strategies, including RAAS blockers, sodium-glucose cotransporter-2 (SGLT2) inhibitors, Glucagon-like Peptide-1 receptor agonists (GLP-1 RA), non-steroidal mineralocorticoid antagonists (ns-MRAs), and their impact on CKD progression are highlighted. Subsequently, combination therapy approaches, the role of biomarkers in personalizing treatments, challenges in implementing new treatments, and future directions in CKD research and clinical management are explored based on comprehensive state-of-the-art CKD care.

## 2. Methods

This review article aimed to identify, evaluate, and synthesize randomized controlled trials (RCTs) assessing pharmaceutical interventions in CKD compared to standard practice or placebo. A systematic search was conducted with Web of Science for papers published between 8 January 2020, and 8 January 2025 (see Appendix B) on CKD treatments, and a citation-per-year cut-off of 10 was applied to limit the number of articles and prioritize the most impactful literature. The primary outcomes of interest were slowed CKD progression and kidney events, while secondary outcomes, when available, included cardiovascular events (Figure 3a). A second, parallel search targeted biomarker-guided interventions from 3 March 2020, to 3 March 2025. Only studies that included patients diagnosed with CKD or diabetic kidney disease (DKD) were eligible for inclusion. Due to a lack of relevant studies in the biomarker-assisted therapy search, no citation limits were applied, and studies regarding biomarker-assisted therapies in other kidney disorders closely related to CKD were also considered (Figure 3b).

The comprehensive search strategy incorporated relevant keywords, synonyms, and appropriate search filters to ensure a robust identification of relevant studies. Only studies published in English were considered. Conference abstracts, articles in press, books, book chapters, commentaries, methodological papers, review articles, editorials, and non-human studies were excluded.

Two authors independently screened the title and abstract. Any discrepancies in inclusion decisions were resolved through consultation with a third author. The screening was facilitated using the web-based software tool Rayyan [16]. Figure 3 and Figure 4 show the PRISMA flow diagrams detailing the study selection process for the CKD treatment and biomarker sections, respectively. Appendix A list all articles that were included for use in this review, for the first and second searches, respectively. While Appendix A provide a comprehensive list of excluded articles with corresponding reasons for exclusion.

**Figure 3 biomolecules-15-00809-f003:**
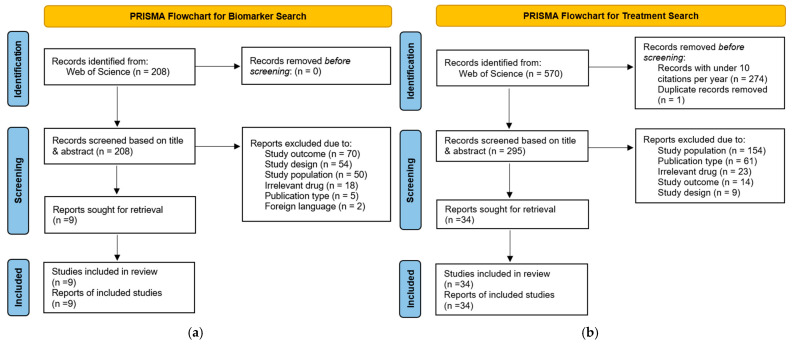
(**a**) PRISMA [17] diagram for treatment search. (**b**) PRISMA diagram for biomarker search. Reasons for exclusion: study population (studies not performed on CKD or DKD patients and all non-human studies); study outcome (studies that did not assess an outcome relevant to this review); publication type (review articles, editorials, commentaries, etc.); study design (in the treatment search, studies that were not randomized controlled trials (RCT), in the biomarker search, studies that were not focused on biomarker-assisted therapy); irrelevant drug (medications that were outside the scope of this review); foreign language (studies that were not in English).

Study characteristics and outcome data were extracted systematically via a standardized form. Extracted variables included (but were not limited to): study title, sample size, demographics, inclusion and exclusion criteria, duration of intervention and follow-up, study design, type of intervention and comparator/control, primary and secondary outcome measures, reported adverse events or complications, primary results, effect size and statistical significance, conclusions, strengths, and limitations.

## 3. Results

### 3.1. Therapeutic Landscape of CKD Management

Over the past five years, CKD management has seen transformative advances with new drug classes that slow disease progression. Traditional therapy for proteinuric CKD centered on blood pressure control and RAAS inhibition. While these measures remain vital, recent landmark trials have expanded the toolkit to include SGLT2 inhibitors and ns-MRAs, among others. Current clinical practice emphasizes a multifactorial approach: controlling risk factors (blood pressure, diabetes, lipids), and using disease-modifying agents to reduce proteinuria and preserve GFR while managing complications (anemia, acidosis, etc.) [6].

#### 3.1.1. Angiotensin-Converting Enzyme (ACE) Inhibitors and Angiotensin II Receptor Blockers (ARBs)

In the 1980s–1990s, blocking the RAAS emerged as a potential CKD therapy. The ACE inhibitor captopril was first shown in 1993 to slow diabetic nephropathy progression in type 1 diabetes, halving the risk of doubling serum creatinine. Subsequent landmark trials in type 2 diabetes (T2D) confirmed the protective effect; the RENAAL trial (2001) found losartan reduced the risk of doubling creatinine by 25% and the risk of end-stage kidney disease (ESKD) by 28% vs. placebo [18], while the IDNT trial (2001) showed irbesartan lowered progression risk by approximately 20–23% compared to placebo or amlodipine [19]. These benefits exceeded those attributable to blood pressure reduction, establishing RAAS blockers as first-line therapy for proteinuric CKD. This class’s mechanism (efferent arteriole dilation resulting in intraglomerular pressure reduction and mitigation of angiotensin II-mediated fibrosis) and trial evidence cemented its role in slowing CKD progression [18,19]. By the 2000s, guidelines universally recommended ACE inhibitors or ARBs for CKD with albuminuria while cautioning against dual RAAS blockade after trials like ONTARGET (2008) showed no added benefit but more adverse effects [20].

The KDIGO 2024 CKD guideline reaffirms RAAS blockade as first-line therapy in patients with CKD and significant albuminuria (A2 or A3), even when estimated (e) GFR falls below 30 mL/min, as long as hyperkalemia is monitored and managed appropriately [7]. This contrasts with previous caution about stopping ACE inhibitors/ARB in advanced CKD and underscores the protective role of RAAS inhibition until very late-stage disease.

#### 3.1.2. Non-Steroidal Mineralocorticoid Receptor Antagonists (ns-MRAs)

MRAs have long been known to attenuate proteinuria and fibrosis, but older steroidal MRAs (spironolactone, eplerenone) saw limited use in CKD due to the risk of hyperkalemia [21]. The development of ns-MRAs marked a renaissance for this class of medications when finerenone, a selective ns-MRA, was proven to reduce albuminuria with a lower hyperkalemia risk than spironolactone [22].

Its pivotal phase III trials in diabetic CKD, FIDELIO-DKD (2020) [23] and FIGARO-DKD (2021) [24], primarily enrolled individuals with T2D and CKD and an eGFR of at least 25 mL/min/1.73 m^2^ with elevated albuminuria despite maximal RAAS blockade, demonstrating significant associated clinical benefits (Table 1).

In FIDELIO-DKD, finerenone lowered the risk of CKD progression or kidney failure by 18% (HR, Hazard Ratio), 0.82; CI (95% Confidence Interval), 0.73–0.93; *p* = 0.001) and reduced cardiovascular events by 14% (HR, 0.86; CI, 0.75–0.99; *p* = 0.03) compared to placebo (on top of standard RAAS inhibitor therapy) [23]. FIGARO-DKD subsequently showed cardiovascular benefits in a broader CKD population of 18% (HR, 0.82; CI, 0.70–0.95; *p* = 0.011) [24]. Analyses from FIGARO-DKD also indicate that finerenone mitigates the risk of initial heart failure hospitalization by approximately 29% (HR, 0.71; CI, 0.56–0.90; *p* = 0.0043) and total hospitalizations for heart failure events by 30% (rate ratio, 0.70; CI, 0.52–0.94) [25,26].

According to FIDELITY (a pooled individual-level analysis conducted across the resulting data of both trials), treatment with finerenone was significantly associated with reductions in cardiovascular events and kidney failure outcomes. For the composite kidney endpoint, the HR was 0.77 (CI, 0.67–0.88; *p* < 0.000), and for the composite cardiovascular endpoint, the HR was 0.86 (CI, 0.78–0.95; *p* = 0.001) [25]. One subgroup analysis showed an even lower HR of approximately 0.78 (CI, 0.57–1.07) for the cardiovascular composite outcome in stage 4 CKD patients [27].

Finerenone elicits a substantial decrease in the urinary albumin-to-creatinine ratio (UACR), with reductions of approximately 30–40% compared to placebo, evident as early as four months post-initiation (*p* < 0.0001) [28,29]. The post hoc mediation analysis confirmed that a 30% reduction in albuminuria accounts for approximately 84% of finerenone’s nephroprotective effect and 37% of its cardiovascular benefit [30]. At the same time, another study assessed that reductions in office systolic blood pressure accounted for a small proportion (13.8% and 12.6%) of the treatment effect of finerenone on the primary kidney composite outcome and the key secondary cardiovascular outcome, respectively [31].

In a pooled analysis of FIDELIO-DKD, FIGARO-DKD, and FINEARTS-HF [32] involving 18,991 participants with cardiovascular-kidney-metabolic syndrome, finerenone demonstrated significant benefits in reducing key clinical outcomes. While the reduction in cardiovascular death (HR, 0.89; CI, 0.78–1.01; *p* = 0.076) was not statistically significant, finerenone significantly lowered all-cause mortality (HR, 0.91; CI, 0.84–0.99; *p* = 0.027), hospitalization for heart failure (HR, 0.83; CI, 0.75–0.92; *p* < 0.001), and the composite kidney outcome (HR, 0.80; CI, 0.72–0.90; *p* < 0.001) [33].

In comparison to steroidal MRAs, such as spironolactone, finerenone has demonstrated a more favorable safety profile in resistant hypertension in moderate-to-advanced CKD. Findings from FIDELITY indicate that although the mean change in systolic blood pressure from baseline to approximately 17 weeks was −7.1 mmHg for finerenone vs. −11.7 mmHg for spironolactone + patiromer and −10.8 mmHg for spironolactone + placebo, hyperkalemia incidence was significantly lower with finerenone at 12%, compared with 35% for spironolactone combined with patiromer and 64% for spironolactone plus placebo [34,35]. Treatment discontinuation due to hyperkalemia occurred in only 0.3% of finerenone-treated participants. In contrast, it was 7% with spironolactone plus patiromer and 23% with spironolactone plus placebo [34,35].

Reflecting these data, the KDIGO 2024 CKD guideline endorses the addition of finerenone (or similar ns-MRAs) in adults with T2D and CKD (stage 1–4 with albuminuria) who remain at high risk despite maximally tolerated RAAS blockade, provided their eGFR is above 25 mL/min/1.73 m^2^, their serum potassium is normal, and eGFR/potassium levels are carefully monitored during treatment [7].

A study conducted in the Netherlands utilized the FINE-CKD model to assess the economic impact of adding finerenone to standard care indicated that it not only improved patient outcomes by extending quality-adjusted life years (QALYs) by 0.20 but also resulted in a reduction in renal and cardiovascular events, culminating in a decrease of €6136 in total lifetime costs per patient [28].

Further investigations will delineate the long-term sustainability of finerenone’s nephroprotective effects, particularly in advanced CKD stages, where progression may attenuate initial benefits [36]. Ongoing real-world registries, such as FINE-REAL, characterize long-term kidney and cardiovascular outcomes, hyperkalemia incidence, and healthcare resource utilization in routine clinical practice [37].

#### 3.1.3. Aldosterone Synthase Inhibitors (ASIs)

The promising results from early-phase studies suggest that ASIs could become a valuable addition to the therapeutic arsenal for CKD. ASIs reduce serum aldosterone levels by approximately 42–66% vs. placebo [38]. This reduction in aldosterone may help slow CKD progression and lower cardiovascular risk, as higher circulating aldosterone levels are associated with increased risk for kidney disease progression [38].

In a phase 2, randomized, controlled trial evaluating BI 690517 (Vicadrostat), adults with albuminuric CKD and a preserved yet declining eGFR were enrolled [39]. Participants were randomly assigned to receive BI 690517 alone or alongside empagliflozin for 14 weeks. The primary endpoint was a proportional change in Urine Albumin-Creatinine Ratio (UACR). Across all dose groups, BI 690517 produced a dose-dependent decline in UACR, with the highest dose reducing albuminuria by nearly 39% relative to placebo (*p* < 0.001) [39]. Concomitant empagliflozin further lessened albuminuria, suggesting complementary mechanisms between mineralocorticoid and SGLT2 pathways.

Although mild-to-moderate hyperkalemia was observed in a minority of participants, most episodes did not necessitate dose adjustment or study withdrawal [39]. The trial’s brevity was noted as a key limitation, emphasizing the need for longer-term studies to determine sustained efficacy, refine hyperkalemia management, and clarify any long-term adrenal effects. The EASi-KIDNEY trial is underway to assess the long-term effects of Vicadrostat on kidney and cardiovascular outcomes in a broader CKD population, aiming to determine if ASIs can provide benefits beyond the current standard of care [40].

#### 3.1.4. Sodium-Glucose Cotransporter-2 (SGLT2) Inhibitors

Developed initially as glucose-lowering drugs, SGLT2 inhibitors have redefined CKD management in the last decade. In 2015, the EMPA-REG OUTCOME trial [41] revealed that empagliflozin improved cardiovascular outcomes in T2D patients and hinted at kidney benefits. Dedicated CKD trials soon confirmed this; the CREDENCE trial (2019) [42] in high-risk diabetic CKD with albuminuria (UACR ≥ 300 mg/g) was stopped early due to the apparent efficacy of canagliflozin; reducing the risk of the primary composite outcome of doubling of serum creatinine, ESKD, or kidney/cardiovascular death by 30% (HR, 0.70; CI, 0.59–0.82; *p* = 0.00001) [42]. Subgroup analyses further revealed that canagliflozin’s protective effects remained consistent across all participants, even those with UACR ≥ 3000 mg/g and those with moderately lower albuminuria [43].

Next, DAPA-CKD (2020) [44] extended these benefits to also include non-diabetic CKD patients; dapagliflozin lowered the risk of a 50% eGFR decline, ESKD, or death by 39% (HR, 0.61; CI, 0.51–0.72, *p* < 0.001) in non-diabetic patients with a number-needed-to-treat of only 19 over 2.4 years. Dapagliflozin slowed chronic eGFR decline (–1.9 vs. –4.0 mL/min/year with placebo) without increased adverse safety signals (*p* < 0.05) [45]. Additionally, even in patients with eGFR < 30 mL/min/1.73 m^2^, dapagliflozin provided significant kidney protection (HR, 0.65; CI, 0.44–0.95; *p* = 0.03) [46]. In a subgroup of patients with biopsy-confirmed focal segmental glomerulosclerosis (FSGS), dapagliflozin was associated with reduced chronic eGFR decline and proteinuria. However, the results did not reach statistical significance, possibly due to the limited sample size (n = 104) [47].

A prespecified analysis focusing on urinary albumin excretion demonstrated that dapagliflozin significantly reduced albuminuria vs. placebo; the geometric mean UACR was lowered by approximately 29.3% (CI –33.1 to –25.2; *p* < 0.0001) [48]. Also, 41% of patients with baseline macroalbuminuria (>300 mg/g) experienced regression to microalbuminuria during the trial, compared to 24% in the placebo group (*p* < 0.001) [48]. Greater reductions in albuminuria correlated with slower eGFR decline (β per log unit UACR change –3.06, CI –5.20 to –0.90; *p* = 0.0056), reinforcing the concept that albuminuria-lowering mediates part of the SGLT2 inhibitors’ protective effect [48].

A pooled analysis of DAPA-CKD and DAPA-HF [49] in individuals without diabetes at baseline (n = 4003) revealed that dapagliflozin reduced the incidence of new-onset T2D by approximately 33% (HR, 0.67; CI, 0.51–0.88, *p* = 0.004), despite no significant effect on mean glycated hemoglobin (HbA_1_c) levels [50].

Most recently, EMPA-KIDNEY (2022) [51] confirmed protection with empagliflozin even in non-albuminuric CKD (approximately 48% of participants had a UACR < 30 mg/g), expanding applicability by exhibiting a 28% relative reduction in the risk of kidney disease progression or cardiovascular death (HR, 0.72; CI, 0.64–0.82; *p* < 0.001) [51]. Secondary analyses also showed that empagliflozin reduced kidney-related hospitalizations (HR, 0.86; CI, 0.75–0.98; *p* = 0.03) [51].

Although focusing mainly on cardiovascular outcomes, a VERTIS trial analysis reported that the SGLT2 inhibitor ertugliflozin significantly reduced an exploratory composite kidney endpoint, including a sustained 40% eGFR decline, dialysis/transplant, or renal death (HR 0.66, CI 0.50–0.88) in patients with T2D and atherosclerotic cardiovascular disease [52]. The dual SGLT1/2 inhibitor sotagliflozin was also evaluated in T2D with stage 3 CKD in a phase three trial, showing improved glycemic control and reduced albuminuria at 26 weeks, though sustained eGFR benefits at 52 weeks remained inconclusive (*p* > 0.05) [53].

A nationwide, multicenter study (J-CKD-DB) found that SGLT2 inhibitors slowed eGFR decline (−0.47 vs. −1.22 mL/min/1.73 m²/year, *p* < 0.001), with an absolute difference of 0.75 mL/min/1.73 m²/year [54]. The risk of ≥50% eGFR decline or ESKD was 60% lower (HR, 0.40; CI, 0.26–0.61; *p* < 0.001) [54]. Benefits were consistent across subgroups, including patients with or without proteinuria and those with rapid eGFR decline [54]. A more substantial effect was observed in patients on ACE inhibitors or ARBs (*p*< 0.05), suggesting synergy with RAAS blockade [54].

SGLT2 inhibitors reduce intraglomerular pressure through tubuloglomerular feedback and mitigate hyperfiltration while exerting a metabolic effect, which may be at least in part responsible for the kidney protective effects, and beyond, e.g., via reduction in body weight and systolic blood pressure [55]. Together, these mechanisms provide synergistic benefits beyond RAAS blockade (Table 2) [56].

Acute eGFR reductions following SGLT2 inhibitor initiation are well documented. Up to 50% of participants in CREDENCE and DAPA-CKD experienced an initial dip exceeding 10% [57,58]. However, this decline is most probably hemodynamic, possibly related to reduced glomerular hyperfiltration mediated by increased sodium chloride transport to the distal tubule and augmented tubulo-glomerular feedback, and not associated with an increased risk of long-term kidney failure, emphasizing the importance of continued therapy despite early fluctuations [59].

A meta-analysis combining seven studies [60] suggests that an initial eGFR dip following SGLT2i initiation is associated with a slower annual eGFR decline (mean difference, 0.64; CI, 0.437–0.843; *p* < 0.001), irrespective of baseline eGFR. Patients with a ≤10% eGFR dip had a reduced risk of major adverse kidney events (MAKE) (HR, 0.915; CI, 0.865–0.967; *p* = 0.002), whereas those with a >10% dip had an increased risk of hyperkalemia (*p* = 0.01). No significant differences were observed between the dipping and non-dipping groups in all-cause mortality (HR, 0.83; CI, 0.589–1.170; *p* = 0.29), heart failure hospitalization (HR, 1.059; CI, 0.574–1.952; *p* = 0.85), or the composite of cardiovascular death and heart failure hospitalization (HR, 0.824; CI, 0.633–1.074; *p* = 0.15). The incidence of serious adverse events, kidney-related adverse events, volume depletion, and SGLT2 inhibitor-related discontinuation remained comparable between groups.

Long-term projections from Markov modeling of DAPA-CKD data suggest substantial benefits. Over 10 years, dapagliflozin could prevent 83 deaths and 51 kidney replacement therapy initiations per 1000 patients [61]. Additionally, the model predicted an incremental gain of 0.56 QALYs per participant over the same timeframe (*p* < 0.01), while healthcare cost-saving analyses estimated that delayed dialysis and reduced heart failure hospitalizations could offset 58–65% of dapagliflozin’s medication costs (*p* < 0.05) [61].

Recognizing these developments, the 2024 KDIGO CKD guideline recommends initiating SGLT2 inhibitors in CKD patients (with or without T2D) with an eGFR ≥ 20 mL/min, cementing their role as a mainstay therapy alongside RAAS blockade [7].

Ongoing research continues to refine surrogate endpoints, while longer-term extension studies and real-world data are essential to determine whether the kidney protective effects of SGLT2 inhibitors persist or even strengthen over time [54,61]. Key implementation challenges include potential cost-related barriers, monitoring volume status, and reluctance to continue therapy after initial eGFR declines. The safety profile of SGLT2 inhibitors remains robust. While volume depletion-related adverse events were slightly more common, large-scale trials found no significant increase in AKI risk or severe volume-related complications [43,45].

#### 3.1.5. Glucagon-like Peptide-1 Receptor Agonists (GLP-1 RA)

GLP-1 RAs, initially introduced for glycemic control in T2D, have gained attention for their kidney and cardiovascular benefits. Liraglutide and semaglutide showed in large outcome trials (LEADER for liraglutide; SUSTAIN-6 and FLOW for semaglutide) that they reduce major adverse cardiovascular events (MACE). However, these trials also collected kidney outcomes as secondary endpoints (Table 3).

In the LEADER trial [62], a significant beneficial effect on eGFR decline was observed in patients with a baseline eGFR < 60 mL/min/1.73 m^2^, with an annual estimated treatment difference of 0.67 mL/min/1.73 m^2^ for liraglutide compared to placebo. Liraglutide reduced the composite kidney outcome by 22% (HR, 0.78; CI, 0.67–0.92) and was driven mainly by less new-onset persistent macroalbuminuria [65]. However, doubling serum creatinine or ESKD rates were similar between groups over around 4 years. Such findings indicated that GLP-1 RAs may slow the progression of DKD, at least in terms of albuminuria [65].

The study’s post hoc mediation analysis revealed that changes in HbA1c explained approximately 25% of the observed kidney benefit with liraglutide (CI, −7.1% to 67.3%), while systolic blood pressure mediated about 9% (CI, 2.8% to 22.7%). Body weight changes also contributed modestly (around 9%) [66]. However, the confidence intervals for these mediation estimates were wide, underscoring the uncertainty on the exact portion of benefit attributable to each factor. Despite these modest mediation effects, new-onset persistent macroalbuminuria emerged as a central driver of the overall kidney benefit with an HR of 0.74 (CI, 0.60–0.91; *p* = 0.004), suggesting that liraglutide’s protective impact extends beyond conventional risk-factor changes and may involve more direct kidney mechanisms [65].

In SUSTAIN 6 [64], the median follow-up was considerably shorter than LEADER (2.1 years), which contributed to fewer kidney events and broader uncertainty for some analyses. Nevertheless, semaglutide reduced the risk of key kidney outcomes with an HR of 0.64 (CI, 0.46–0.88; *p* = 0.005). Among those with a baseline eGFR < 60 mL/min/1.73 m^2^, the annual estimated treatment difference favoring semaglutide was 1.62 mL/min/1.73 m^2^ compared to placebo, indicating a notable slowing of kidney function loss [66].

Semaglutide’s observed kidney benefits in SUSTAIN 6 were modestly explained by reductions in systolic blood pressure (around 22%) and HbA_1_c (around 26%), with body weight contributing little or no additional mediation [66]. It is important to note, however, that most GLP-1 RA trials have predominantly enrolled overweight or obese populations, and dedicated trials in normal-weight individuals or in patients who do not lose weight are lacking. Nevertheless, additional data from extended trials or pooled analyses are needed to clarify which mechanistic pathways underlie semaglutide’s benefits.

In the FLOW trial [63], semaglutide again demonstrated significant kidney and cardiovascular benefits over a median follow-up of 3.4 years. The primary endpoint (a composite of new-onset kidney failure, ≥50% decline in eGFR, or death from kidney-related or cardiovascular causes) occurred significantly less often in the semaglutide group, corresponding to a 24% reduction in relative risk (HR, 0.76; CI, 0.66–0.88; *p* = 0.0003). Additionally, semaglutide slowed the annual decline in eGFR by 1.16 mL/min/1.73 m^2^ (*p* < 0.001). Cardiovascular benefits were also evident, with semaglutide reducing the risk of MACE by 18% (HR, 0.82; CI, 0.68–0.98; *p* = 0.029) and all-cause mortality by 20% (HR, 0.80; CI, 0.67–0.95, *p* = 0.01). Safety assessments revealed a favorable overall safety profile for semaglutide, with fewer serious adverse events than placebo (49.6% vs. 53.8%) [67]. However, permanent treatment discontinuation rates were higher, primarily due to gastrointestinal side effects and, in some cases, retinal complications [67].

While semaglutide slowed eGFR decline in a high-risk CKD population, its generalizability to lower-risk or dialysis-dependent groups remains uncertain. The event-driven design of FLOW provided robust evidence for primary endpoint assessment but left some long-term questions unresolved, particularly regarding sustained kidney protection and rare but significant adverse effects. Future research should focus on validating these kidney-protective benefits across diverse populations, evaluating dual or triple-agent regimens, and defining the optimal role of semaglutide in CKD treatment.

In 2022, KDIGO’s diabetes in CKD guidelines recommended long-acting GLP-1 RAs for T2D patients with CKD who need additional glycemic or weight control despite metformin and an SGLT2 inhibitor [68]. The KDIGO 2024 CKD treatment guideline aligns with its earlier diabetes guidance by recommending long-acting GLP-1 RA in T2D patients with CKD who are obese and require further glycemic control or cardioprotection beyond standard therapy (metformin and SGLT2 inhibitors) [7].

#### 3.1.6. Endothelin Receptor Antagonists (ERA)

Endothelin-1, a potent vasoconstrictor and promoter of fibrosis, became a therapeutic target in proteinuric CKD after preclinical studies showed it contributes to glomerular injury. Early attempts in the 2000s with ERAs like avosentan showed marked albuminuria reductions (around 44–49% decrease in UACR) but at the cost of significant fluid retention and heart failure [69]. The phase 3 ASCEND trial of avosentan in diabetic nephropathy had to be terminated early (median 4 months) due to excess cardiovascular events (edema and congestive heart failure), and it ultimately showed no improvement in hard kidney outcomes [69]. Researchers then pursued selective endothelin A antagonists at lower doses, and the SONAR trial (2019) [70] of atrasentan in DKD used an “enrichment” design to mitigate risk: only patients who responded to atrasentan and tolerated it during a run-in were randomized. SONAR demonstrated a significant 35% reduction in doubling of creatinine or progression to ESKD with atrasentan (HR, 0.65; CI, 0.49–0.88; *p* = 0.0047) [70]. However, edema and congestive heart failure risks were higher with atrasentan (though not statistically significant for heart failure [70]. These mixed results led to ERAs not being considered for standard therapy in CKD.

A newer dual endothelin and angiotensin II receptor antagonist (sparsentan) was tested in IgA nephropathy in the PROTECT trial (2023) [71], showing superior proteinuria reduction vs. an ARB alone and leading to FDA approval for this drug in IgA nephropathy [72]. At 36 weeks, sparsentan reduced proteinuria by 49.8% vs. 15.1% with irbesartan, representing a 41% relative reduction (Least Squares Mean Ratio, 0.59; CI, 0.51–0.69; *p* < 0.0001). Over 110 weeks, sparsentan’s chronic eGFR decline was −2.7 mL/min/1.73 m^2^/year vs. −3.8 mL/min/1.73 m^2^/year with irbesartan, a difference of +1.1 mL/min/1.73 m^2^/year (CI 0.1–2.1; *p* = 0.037) [71]. The composite kidney failure endpoint occurred in 9% of sparsentan patients compared with 13% for irbesartan (relative risk, 0.7; CI, 0.4–1.2) [71]. This indicates a niche role for endothelin pathway blockade in specific glomerular diseases, and sparsentan’s role in broader CKD is currently under study. Overall, ERAs represent a promising but cautious avenue; they may be reserved for refractory proteinuria and are pending further trials to balance efficacy and safety.

### 3.2. Combination Therapy Approaches

#### 3.2.1. Latest Evidence

Combination pharmacotherapy in CKD has garnered growing attention, as multiple complementary pathways (including RAAS overactivation, glomerular hypertension, and metabolic stress) drive progressive kidney dysfunction. Newer agent classes, particularly SGLT2 inhibitors, ns-MRAs, GLP-1 RAs, and, to a lesser degree, ERAs, demonstrate substantial kidney and cardiovascular protective effects. Consequently, efforts are initiated towards testing dual and even triple combination regimens, aiming to maximize kidney protection while maintaining an acceptable safety profile [73,74,75].

Despite the potential benefits of early multi-target intervention, safety concerns such as hyperkalemia and fluid retention must be carefully managed. Elevated potassium remains the key side effect of MRAs; however, combining an MRA with an SGLT2 inhibitor may reduce that risk [76].

A crossover trial examining dapagliflozin, eplerenone, and their combination over four weeks in 46 patients showed a mean percentage change in UACR of −19.6% (CI, −34.3 to −1.5) for dapagliflozin (*p* < 0.05), −33.7% (CI, −46.1 to −18.5) for eplerenone (*p* < 0.01), and −53.0% (CI, −61.7 to −42.4; *p* < 0.001) for the combination [76]. Hyperkalemia occurred in 17.4% of patients receiving eplerenone alone, compared with 0% for dapagliflozin alone and 4.3% on combined therapy, suggesting that SGLT2 inhibitors may mitigate the potassium elevation typical of MRA therapy [76]. Meanwhile, in DAPA-CKD, dapagliflozin retained its kidney-protective effects even in patients receiving MRAs, with no substantial increase in hyperkalemia (*p* > 0.05) [77].

A study on the FIDELITY dataset demonstrated that finerenone provided comparable risk reductions in kidney and cardiovascular outcomes in patients on SGLT2 inhibitor treatment compared to those not, with preliminary signals suggesting a possible attenuation of hyperkalemia among concurrent SGLT2 inhibitor users [74]. Subgroup analyses of the finerenone trials also demonstrated that finerenone’s kidney-protective and cardiovascular benefits persist regardless of the concurrent use of GLP-1 RA or SGLT2 inhibitors [78]. In patients receiving GLP-1 RAs, finerenone showed no significant interaction concerns, maintaining an acceptable safety profile and cardiorenal efficacy despite this subgroup’s relatively small sample size (n = 394) [78].

The CONFIDENCE trial, a phase 2 study involving 807 adults with stage 2–3 CKD and T2D, aims to evaluate the effectiveness of finerenone combined with an SGLT2 inhibitor over six months. The primary endpoint is the relative change in UACR among those receiving dual therapy vs. monotherapy. If the study meets its primary objective, showing superior albuminuria reduction with combination therapy, it could accelerate phase 3 testing of early combined MRA–SGLT2i initiation [73].

The SONAR study [79] suggests combining SGLT2 inhibitors with atrasentan may enhance kidney protection by reducing albuminuria and mitigating fluid retention. The combination resulted in a net weight reduction of –1.2 kg (CI, –2.3 to –0.1 kg; *p* = 0.03) vs. atrasentan alone, indicating less fluid accumulation. Initiating SGLT2 inhibitor therapy during the atrasentan run-in phase led to an additional 27.6% UACR decline (CI, 3.6−45.6%; *p* = 0.028), supporting their complementary nephroprotective effects. Although limited to six weeks, these findings suggest that SGLT2 inhibitors’ natriuretic action may offset atrasentan-induced fluid retention. A fully powered randomized trial is needed to confirm the long-term efficacy and safety of this combination.

The FLOW trial also explored the potential synergy between semaglutide and SGLT2 inhibitors [63,67]. Despite only 15.6% of participants being on SGLT2 inhibitors at baseline, semaglutide’s observed kidney and cardiovascular benefits remained consistent regardless of SGLT2 inhibitor use, with no evidence of outcome heterogeneity. These findings suggest that semaglutide’s protective effects were independent of background SGLT2 inhibitor therapy and that both treatments may offer complementary benefits for kidney and heart health in high-risk individuals. Notably, no additional safety concerns emerged from the combination of semaglutide and SGLT2 inhibitors. Despite these promising findings, the study had limited statistical power to assess differences within key subgroups, particularly non-White populations, and the relatively low SGLT2 inhibitor usage at baseline restricts insights into combination therapy effects.

Regarding a potential combination of ERAs and SGLT2 inhibitors, the ZENITH-CKD trial [80] investigated zibotentan combined with dapagliflozin. The study included 447 patients and found a decrease in UACR at week 12 of −33.7% (*p* < 0.0001) for the zibotentan 1.5 mg plus dapagliflozin group and −27.0% (*p* = 0.0022) for the zibotentan 0.25 mg plus dapagliflozin group, compared to dapagliflozin plus placebo. However, fluid retention occurred in 18% of patients on zibotentan 1.5 mg plus dapagliflozin, vs. 8% with dapagliflozin alone. These data highlight the trade-off between enhanced albuminuria reduction and mild increases in fluid retention.

Real-world feasibility, cost, and patient adherence also remain critical considerations. Larger-scale trials such as PRECIDENTD and ongoing observational studies will provide further insight into the efficacy and practicality of multi-agent therapy in CKD–T2D populations. The rationale for combining SGLT2 inhibitors, MRAs, GLP-1RAs, and potentially ERAs may gain support with promising data suggesting that such combinations can significantly slow CKD progression and reduce cardiovascular risk.

#### 3.2.2. Advantages & Disadvantages of Combination Therapy

Large trials have demonstrated the clinical value of fixed-dose combination therapy in the context of cardiovascular disease [81,82,83], yet their use in CKD has historically been limited by the lack of multiple effective disease-modifying therapies. However, recent advances such as the emergence of SGLT2 inhibitors as foundational therapy alongside RAAS blockade have shifted this landscape. These developments now make the implementation of polypill strategies in CKD a feasible and promising avenue.

The rationale for combination therapy in CKD is multifaceted. On the one hand, multiple pharmacologic agents targeting distinct pathophysiological pathways can provide synergistic or at least complementary benefits, potentially improving outcomes, particularly in complex presentations that involve concomitant hypertension, metabolic dysregulations, and proteinuria. For instance, combining ARBs with an SGLT2 inhibitor could yield added benefits by addressing glomerular hypertension and tubulo-glomerular feedback.

On the other hand, while combining multiple novel agents may confer enhanced kidney protection, it also accentuates cost and polypharmacy-related risks. Also, there are inherent limitations to an overly generalized or “one-size-fits-all” combination strategy. CKD progression exhibits considerable interindividual variability, and therapeutic responses can differ substantially based on genetic, metabolic, or proteomic factors. A standardized combination approach risks overlooking patient-specific nuances, such as susceptibility to hyperkalemia, volume status, or coexisting conditions. Finally, this approach imposes higher costs that rigorous cost-effectiveness analyses must justify until they are adopted as standard care. Thus, while combination therapies can streamline regimens and enhance outcomes for carefully selected patients, they require prudent selection and close monitoring to ensure that compounding side effects and mechanistic redundancies do not undermine the clinical advantage.

When combination regimens are structured appropriately, they can reduce disease progression and potentially simplify patient management. A single-pill combination, for example, may bolster adherence by reducing pill burden, an especially relevant consideration in CKD patients who often face polypharmacy. Streamlining treatment can also improve patient satisfaction and lead to more consistent dosing, factors that translate to better long-term outcomes. Patients who are concurrently grappling with edema, hypertension, diabetes, and dyslipidemia are particularly likely to benefit from a carefully selected combination of interventions that address multiple targets in parallel.

### 3.3. Biomarker-Guided Interventions

The growing interest in biomarker-guided treatments aims to refine therapeutic decision-making by identifying molecular and clinical markers that predict disease progression and treatment response (Figure 4). Such an approach is by now considered routine in oncology, where an armamentarium of drugs aimed at specific targets is available [84]. Recent work has focused on the role of urinary peptides, proteomic signatures, and inflammatory mediators in guiding therapy across various CKD etiologies, including DKD, membranous nephropathy, and autoimmune tubulointerstitial nephritis (TIN).

#### 3.3.1. Urinary Peptide Signatures for Predicting Treatment Response

Several studies indicated that the response to specific interventions can be predicted based on urinary peptides; albuminuria response to spironolactone [85], RAAS blockade in DKD [86] and response to immunosuppressive treatment in IgAN [87] was reported as significantly associated with specific urinary peptides. These data sparked the retrospective in silico analysis of 5585 individuals to evaluate three urinary peptide classifiers (HF2, CAD160, and CKD273) for predicting heart failure, coronary artery disease, and CKD events [88]. The study demonstrated strong correlations between higher classifier scores and increased event rates; an adjusted version of each model accounting for age, blood pressure, sex, body mass index, and eGFR was also tested (Table 4).

To explore therapeutic implications, the researchers applied in silico modeling to predict the effects of ARBs, MRAs, SGLT2 inhibitors, GLP-1RA, dipeptidyl peptidase-4 inhibitors (DPP4i), olive oil supplementation, and exercise. They recalculated classifier scores after adjusting peptide profiles according to known intervention-induced shifts. SGLT2i and ARBs demonstrated substantial predicted benefits in patients with high HF2 or CKD273 risk, aligning with clinical trial data supporting their cardioprotective and kidney protective effects. In contrast, GLP1-RA and olive oil exhibited more modest benefits, particularly in lower-risk patients. While promising, the authors emphasize that these findings require prospective validation to confirm the utility of urinary proteomics in guiding personalized therapy selection.

Following up, the same group investigated the prediction of response to different combinations of six types of intervention in 935 patients with CKD. Based on regression, the authors transformed the CKD273 score until a 50% chance of MAKE was predicted to present an apparent patient-relevant endpoint. Investigating which treatment would delay time until 50% MAKE most, the optimal drug (combination) was chosen. Surprisingly, in >80% of cases, combining all possible treatments was not the optimal approach. The authors correctly indicate that the study has several shortcomings; it is based only on in silico simulation, nevertheless, such an approach shows promise for guiding CKD interventions in real-world scenarios.

#### 3.3.2. Immune System Related Biomarkers

TNF receptors and kidney injury molecule-1 (KIM-1) have gained particular traction in identifying patients most likely to benefit from SGLT2 inhibition. In the CANVAS trial, each doubling of baseline TNFR-1, TNFR-2, and KIM-1 was linked to a markedly higher risk of adverse kidney outcomes, with HRs of 3.7, 2.7, and 1.5, respectively [89]. Compared with placebo, canagliflozin modestly reduced TNFR-1 (2.8%) and TNFR-2 (1.9%) levels, producing a more substantial 26.7% decrease in KIM-1. Notably, within the canagliflozin group, every 10% reduction in TNFR-1 and TNFR-2 at one year was independently associated with a lower hazard of kidney disease progression. These findings suggest that early decreases in TNFR-1 and TNFR-2 may serve as valuable pharmacodynamic markers of kidney-specific therapeutic benefit [89].

In a single-center retrospective analysis of 62 individuals diagnosed with TIN, including 30 with autoimmune TIN, elevated serum soluble interleukin-2 receptor (sIL-2R) levels were shown to predict kidney functional recovery at three months [90]. Multivariate analysis revealed a significant positive association between higher sIL-2R levels and improved eGFR (β = 1.102; *p* < 0.001). ROC analysis yielded an AUC of 0.805, with a sensitivity of 0.90 and a specificity of 0.55 at a cut-off value of 1182 U/mL. These findings highlight sIL-2R as a potentially valuable biomarker for identifying patients who may benefit most from aggressive immunosuppressive therapy, particularly glucocorticoids.

A recent study examined the role of TGF-β in predicting cyclophosphamide (CYC) therapy outcomes among pediatric patients with steroid-resistant nephrotic syndrome [91]. In a cohort of 88 children, baseline serum TGF-β levels were measured before CYC initiation. Each one-unit increase in TGF-β was associated with an adjusted odds ratio of 1.051 (95% CI 1.007–1.097; *p* = 0.022) for failure to achieve remission. Though limited in size, these findings suggest that TGF-β profibrotic activity may drive ongoing renal fibrosis, thereby diminishing the efficacy of alkylating agents in this subgroup.

Another study evaluated serum rituximab concentrations in 68 patients with primary membranous nephropathy (pMN) to determine their prognostic value for treatment response [92]. Undetectable rituximab levels at three months post-administration were strongly associated with treatment failure, defined as a lack of remission at both six and twelve months. Moreover, participants with baseline serum albumin below 22.5 g/L were likelier to exhibit undetectable rituximab levels by the third month, potentially due to severe nephrotic-range proteinuria leading to urinary loss of rituximab.

Finally, a study investigated the prognostic utility of the neutrophil-to-lymphocyte ratio (NLR) and platelet-to-lymphocyte ratio (PLR) in 50 pediatric nephrotic syndrome patients undergoing steroid therapy found no significant changes in NLR or PLR following steroid treatment, nor did these indices predict steroid responsiveness [93].

#### 3.3.3. Additional Predictors of Kidney Outcomes

While the majority of CKD cases result from modifiable lifestyle and environmental factors, advances in genomics have shed light on genetic factors that confer CKD susceptibility. For instance, variants in the APOL1 gene (common in individuals of African ancestry) significantly increase the risk of CKD and kidney failure. In recent years, APOL1-mediated kidney disease (such as FSGS in carriers of two APOL1 risk alleles) has become a focus of precision medicine research [94]. This has resulted in the development of an APOL1 inhibitor (inaxaplin), a novel small molecule directly targeting the APOL1 protein’s pathogenic effects. In a 2023 trial, inaxaplin significantly reduced proteinuria by approximately 47% in patients with APOL1-associated proteinuric CKD, demonstrating the potential of genotype-targeted treatment [94].

When analyzing 100 biopsy-proven proliferative lupus nephritis (PLN) cases [95], baseline proteinuria levels below 1.5 g/day were associated with a shorter time to complete response, while proteinuria levels above 0.8 g/day at twelve months correlated with higher flare rates. Consistency across multiple time points highlights proteinuria as a key prognostic indicator in lupus nephritis. Histopathological analysis revealed that interstitial fibrosis/tubular atrophy greater than 25% predicted progression to stage 3–4 CKD or ESKD.

In a study of adults with T2DM and proteinuric CKD, baseline 24 h urinary protein (uProt), eGFR, and BMI were strongly predictive of a 30% reduction in uProt after initiating SGLT2 inhibitor therapy. Notably, patients who achieved ≥ 30% proteinuria reduction (responders) experienced a significant decrease in eGFR (−10.43%) compared to non-responders (*p* = 0.017), highlighting differential response patterns across distinct DKD phenotypes [96].

In the SONAR trial, those with ≥60% UACR reduction on atrasentan (after pre-enrichment) showed a 75% lower hazard for doubling of serum creatinine or ESKD (HR = 0.25; CI 0.11–0.59) compared with those who failed to lower UACR by 15%. However, atrasentan’s benefits remained consistent across UACR response strata, reducing kidney risk and slowing eGFR decline regardless of early albuminuria changes. Therefore, while substantial initial drops in albuminuria correlated with better kidney outcomes, they did not robustly predict who derives the greatest long-term benefit, emphasizing the need for additional biomarkers or approaches to distinguish true responders [97].

A summary of the studies found in our systematic search are available in Table 5.

#### 3.3.4. Future Directions and Clinical Implications

CKD presents with diverse etiologies and progression patterns, making a standardized therapeutic approach inadequate for many patients. Personalized treatment strategies leverage biomarker data to refine therapeutic decisions, optimizing efficacy and safety while reducing healthcare costs. Personalized strategies also reduce adverse effects by identifying patients at risk for drug toxicity or non-responsiveness by identifying individuals most likely to benefit from specific therapies and doses. Large-scale, prospective, multicenter trials are necessary to confirm the clinical utility of biomarkers in predicting treatment response. Although further prospective trials and large-scale validation studies are warranted, the findings presented in this review mark a transition toward precision nephrology, moving away from standardized treatment models toward truly individualized patient care.

Despite its promise, integrating biomarker-driven nephrology into routine clinical practice presents several obstacles. Cost and accessibility remain major barriers, as advanced assays require specialized infrastructure and expertise, limiting their availability. Logistical challenges also arise when integrating biomarkers into standard CKD workflows. Clinician and laboratory personnel training, as well as standardized protocols, are necessary to ensure effective implementation. However, considering the extremely high costs of CKD and the highly positive impact such a strategy has been shown to have in oncology, it is to be expected that existing hurdles will soon be overcome, in the interest of patients, but also due to predicted major long-term cost benefits.

Regulatory and reimbursement issues further hinder adoption. Health insurance providers and policymakers demand high-quality evidence demonstrating that biomarker-driven strategies improve outcomes and are cost-effective. Even validated biomarkers may remain in a non-reimbursed category, limiting their clinical use. Addressing these challenges will require collaboration among researchers, clinicians, and policymakers. Standardized reporting, cost-effectiveness studies, and consensus guidelines can accelerate the transition from research to clinical application, ultimately making personalized nephrology a reality for CKD patients worldwide.

While there is growing interest in leveraging additional biomarkers (beyond eGFR and albuminuria) to enable more precise risk stratification and individualized therapy, current KDIGO guidelines do not yet formally endorse novel biomarkers for routine clinical decision-making. Nonetheless, the guidelines acknowledge the potential for biomarker-guided interventions as emerging evidence solidifies their clinical utility.

## 4. Discussion

The therapeutic landscape for CKD has evolved substantially, propelled by evidence that multi-agent regimens can confer synergistic kidney protective effects. However, despite clear mechanistic rationales, thorough evidence for their cost-effectiveness and real-world impact remains fragmented. Most contemporary trials benchmark novel agents against ACE inhibitor/ARB monotherapy rather than conduct head-to-head comparisons with now-established combination therapies, making quantifying incremental or synergistic benefits difficult. This gap presents a key challenge to clinicians seeking to tailor interventions, especially given the complexities of polypharmacy, ranging from higher financial outlays to increased risk of adverse events.

A potential solution involves biomarker-informed management, which aligns therapies with molecular risk profiles and predicted drug responsiveness. By identifying the subset of patients most likely to benefit from multi-drug interventions, biomarker guidance can minimize the clinical and economic burden on those unlikely to respond. In this context, the CKD273 classifier has gained significant attention. This test employs an SVM trained on 273 urinary peptides to predict progression risk in diabetic CKD using three pivotal cut-offs [98,99,100,101]. Values above 0.154 indicate early progression risk, those exceeding 0.343 point to advanced CKD and increased mortality over six years, and scores beyond 0.55 strongly correlate with ESKD or death. In the PRIORITY trial, a CKD273 score of 0.154 identified 12.1% of normo-albuminuric T2D patients with a 2.48-fold (CI: 1.51–4.08; *p* < 0.001) increased risk to develop albuminuria, after adjusting for baseline clinical factors [102]. This demonstrates that the proteomic classifier CKD273 enables identifying patients who may benefit from earlier interventionCKD273 also exhibits an inverse relationship with eGFR (R = −0.64) [101], while often surpassing albuminuria and the Kidney Failure Risk Equation (KFRE) in early CKD stages [103].

Despite a single-use cost of approximately €850, it has demonstrated cost-effectiveness in high-risk populations by triggering earlier interventions that improve outcomes by an estimated 0.13 QALYs [104]. Moreover, emerging data suggest that one CKD273 measurement can facilitate in silico simulation of different pharmacologic regimens by mapping their peptide-altering effects [86,88,105]. Such insights could guide more personalized therapy choices, circumventing a trial-and-error approach that is both costly and time-intensive (Figure 5).

Meanwhile, evidence continues to endorse screening and early intervention as cornerstones of effective CKD management. A systematic review of 21 studies concluded that, although universal CKD screening in unselected populations yields mixed results, targeted screening among high-risk groups (individuals with diabetes, hypertension, advanced age, or specific ethnic backgrounds) proves cost-effective [106]. Integrating molecular biomarker assays such as CKD273 may further fill this gap by detecting subtle pathophysiological shifts that precede overt changes in clinical parameters, leading to earlier diagnoses in populations at risk for rapid CKD progression [107].

Economic considerations inevitably underscore any treatment strategy. Table 6 outlines the approximate monthly and annual costs of the primary interventions mentioned in Germany. Generic therapies such as ACE inhibitors or ARBs remain inexpensive, whereas newer agents, ns-MRAs and GLP-1RA, introduce higher costs. With dialysis costing upwards of €130 per day, the stakes for preventing or deferring kidney replacement therapy are indisputably high.

An attractive hybrid paradigm merges combination therapy with biomarker-guided escalation or de-escalation. Physicians could initiate a polypill in patients with high molecular risk signatures and then intensify, modify, or discontinue specific components based on biomarker assessments. For example, in early-stage CKD a streamlined yet unguided fixed-dose polypill built around an SGLT2 inhibitor plus an ARB may offer practical benefits by delivering broad, guideline conforming protection (upon validation in a properly powered clinical trial). Conversely, in advanced CKD where patients are at massively increased risk of developing ESKD and/or experiencing cardiovascular events, personalized approaches appear to be the superior choice, as these are expected to lead to better outcomes and spare patients’ unnecessary polypharmacy or complications.

Though conceptually compelling, large-scale RCTs are indispensable for evaluating whether the added biomarker expense justifies improved clinical outcomes and offsets the cost of multi-agent therapy. Ideally, future trials would integrate biomarker protocols from the outset, randomizing participants to usual care vs. biomarker-guided therapy arms, with concurrent pharmacoeconomic analyses.

These perspectives illustrate that both multi-drug therapy and biomarker-guided management address longstanding challenges in CKD care. Combination therapy can exploit pharmacologic synergies to curtail disease progression more robustly than monotherapy, but its broad application raises questions about cost, safety, and patient variability. Biomarker-driven tailoring can mitigate these pitfalls, but evidence supporting its seamless integration into standard practice remains limited. A polypill-plus-biomarker strategy could theoretically balance the benefits of multi-agent coverage with the need to avoid overtreatment, yet robust empirical data are lacking.

Although we employed a systematic search in our methodology, several factors temper the certainty of our conclusions. First, both searches were restricted to Web of Science and the treatments search was filtered for studies with ≥10 citations year^−1^, which may have excluded some possibly relevant literature. Second, we analyzed published summary data and did not explore individual-participant data to address heterogeneity or perform subgroup meta-analysis as it was beyond the scope of this manuscript. Third, risk-of-bias analysis relied on information reported by trialists, which may under-represent selective-reporting or publication biases.

Moving forward, the research agenda should prioritize prospective, head-to-head trials that compare novel multi-agent regimens under real-world conditions, embedding biomarker testing within the initial and follow-up assessments. Parallel economic evaluations can then quantify whether biomarker-driven de-escalation yields sufficient cost offsets to justify investments in diagnostic tools like CKD273. Clinicians, researchers, and payers can forge consensus on optimal CKD treatment algorithms only through large-scale, controlled evaluations replicating routine clinical settings. Such efforts promise to solidify the role of precision medicine in CKD by delivering improved clinical outcomes and reducing healthcare expenditures, which are crucial dual objectives in an era of limited resources.

## 5. Conclusions

CKD management is advancing rapidly, driven by the swift development of combination treatments and innovative approaches for personalizing interventions using biomarkers. Future clinical trials should compare new therapies against the most recently validated regimens to further refine these strategies, generating robust data on optimal combination approaches. Moreover, biomarker-assisted care shows excellent potential in the personalized management of CKD.

## Figures and Tables

**Figure 1 biomolecules-15-00809-f001:**
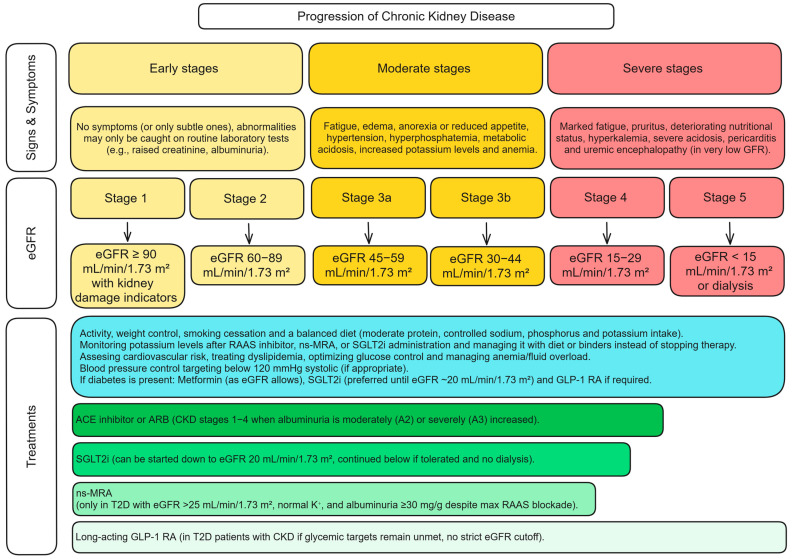
Summary of CKD progression, diagnosis, and respective treatment options [7]. Albuminuria categories (urine albumin-to-creatinine ratio, mg/g): A1: <30 (normal to mildly increased), A2: 30–300 (moderately increased), A3: >300 (severely increased). Abbreviations: ACE—Angiotensin-Converting Enzyme; ARBs—Angiotensin II Receptor Blockers; CKD—Chronic Kidney Disease; eGFR—Estimated Glomerular Filtration Rate (mL/min/1.73 m^2^); GLP-1 RA—Glucagon-Like Peptide-1 Receptor Agonist; K^+^—Potassium; ns-MRA—Non-steroidal Mineralocorticoid Receptor Antagonist; RAAS—Renin-Angiotensin-Aldosterone System; SGLT2i—Sodium-Glucose Cotransporter-2 Inhibitors; T2D—Type 2 Diabetes.

**Figure 2 biomolecules-15-00809-f002:**
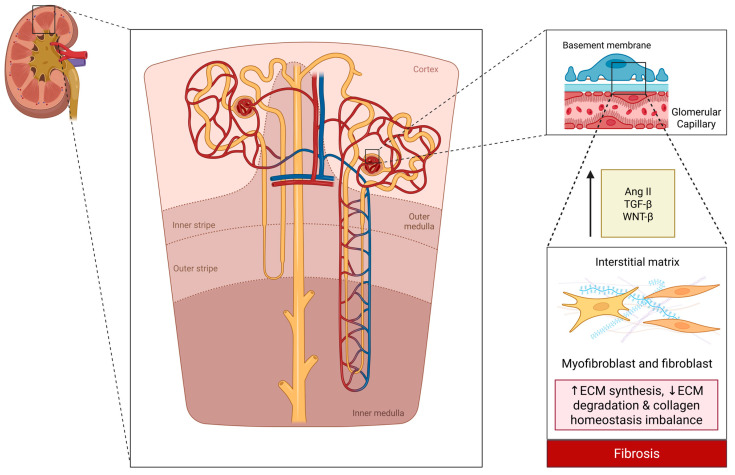
Schematic representation of renal architecture and fibrotic pathways in chronic kidney disease. This schematic illustrates a kidney cross-section with major zones (cortex, medulla) and a representative nephron. An inset highlights the glomerular filtration barrier (endothelium, basement membrane, podocytes), while the lower section depicts key fibrotic signals (Ang II, TGF-β, WNT/β-catenin) that promote extracellular matrix imbalance and fibrosis. Created in BioRender. Biglari, S. https://BioRender.com/rrgof5o (accessed on 17 April 2025). Abbreviations: Ang II—Angiotensin II; ECM—Extracellular Matrix; MMP—Matrix Metalloproteinase; TGF-β—Transforming Growth Factor Beta; WNT-β—Wnt/β-catenin signaling.

**Figure 4 biomolecules-15-00809-f004:**
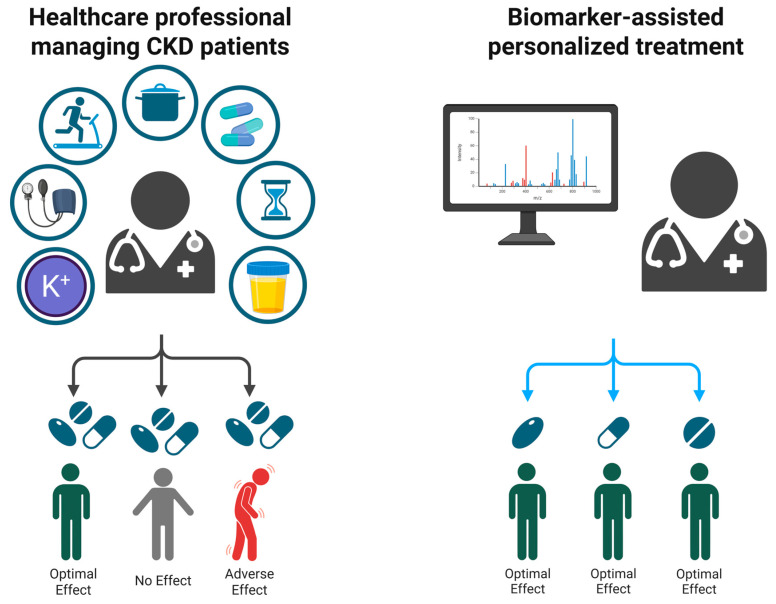
Enhancing CKD treatment via biomarker stratification. This figure contrasts a conventional, trial-and-error approach to CKD management with a biomarker-driven, precision strategy. In the traditional model, identical treatments can produce variable outcomes, ranging from successful responses to adverse effects. By incorporating patient-specific biomarker data, therapies are tailored to individual molecular profiles, optimizing clinical results. Created in BioRender. Biglari, S. https://BioRender.com/7kujt1p (accessed on 17 April 2025).

**Figure 5 biomolecules-15-00809-f005:**
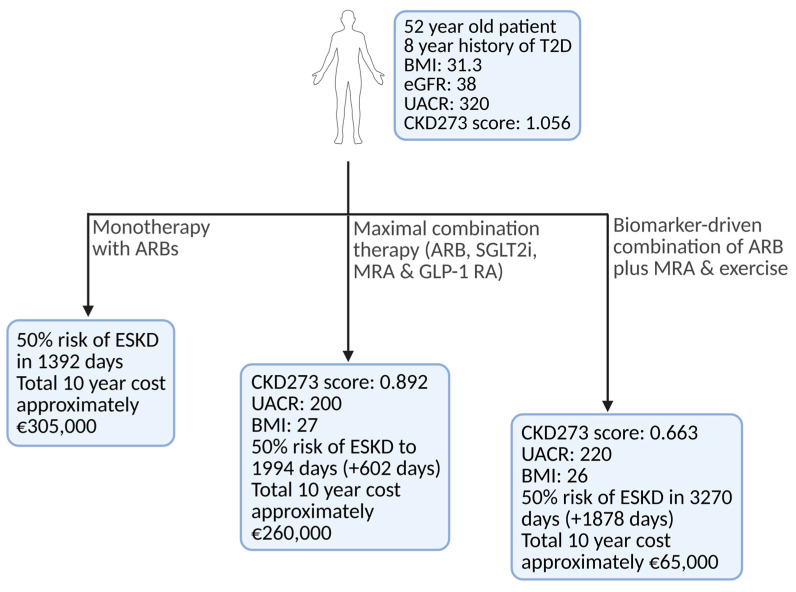
Hypothetical patient receiving CKD treatment with three distinct potential approaches. Over a 10-year horizon, the expanded four-drug regimen is projected to cost around €260,000 (encompassing both medication and dialysis), whereas a more streamlined dual-therapy approach using an ARB, an MRA (and exercise) would total only about €65,000. In stark contrast, ARB monotherapy, despite its low monthly cost, hastens the need for dialysis and ultimately drives total 10-year expenditures to approximately €305,000. Although these effects have currently only been demonstrated via in silico modeling [105], they clearly illustrate the profound clinical and economic impact of carefully tailored biomarker-driven combination therapies. Created in BioRender. Biglari, S. https://BioRender.com/1hbou8k (accessed on 17 April 2025). Abbreviations: ARB—Angiotensin II Receptor Blocker; BMI—Body Mass Index; CKD273—Chronic Kidney Disease classifier with 273 peptides; eGFR—Estimated Glomerular Filtration Rate; ESKD—End-Stage Kidney Disease; GLP-1RA—Glucagon-Like Peptide-1 Receptor Agonist; MRA—Mineralocorticoid Receptor Antagonist; SGLT2i—Sodium-Glucose Cotransporter-2 Inhibitor; T2D—Type 2 Diabetes; UACR—Urine Albumin-to-Creatinine Ratio.

**Table 1 biomolecules-15-00809-t001:** Summary of finerenone trials.

Trial	Patients	Intervention	Key Outcome	Notable Point
FIDELIO-DKD (NCT02540993) [23]	5734 individuals with T2D & CKD (eGFR 25–60 mL/min/1.73 m^2^ and UACR 30–300 mg/g, or eGFR 25–75 mL/min/1.73 m^2^ and UACR 300–5000 mg/g)	Finerenone vs. placebo (on top of RAAS blockade)	18% risk reduction in kidney failure, ≥40% decline in eGFR, or renal death	Significant renal benefits, especially in patients with CKD and diabetes
FIGARO-DKD (NCT02545049) [24]	7437 individuals with T2D & CKD (eGFR 25–90 mL/min/1.73 m^2^ and UACR 30–300 mg/g, or eGFR > 60 mL/min/1.73 m^2^ and UACR 300–5000 mg/g)	Finerenone vs. placebo (on top of RAAS blockade)	13% risk reduction in CV death, non-fatal MI, non-fatal stroke, or HF hospitalization	Significant kidney & CV benefits, especially in patients with CKD and diabetes

Abbreviations: CKD—Chronic Kidney Disease; CV—Cardiovascular; eGFR—Estimated Glomerular Filtration Rate (mL/min/1.73 m^2^); HF—Heart Failure; MI—Myocardial Infarction; RAAS—Renin-Angiotensin-Aldosterone System; T2D—Type 2 Diabetes; UACR—Urine Albumin-to-Creatinine Ratio (mg/g).

**Table 2 biomolecules-15-00809-t002:** Summary of SGLT2 inhibitor trials.

Trial	Patients	Intervention	Key Outcome	Notable Point
CREDENCE (NCT02065791) [42]	4401 individuals with T2D and CKD (eGFR 30–90 mL/min/1.73 m^2^, UACR > 300 mg/g)	Canagliflozin vs. placebo (in addition to standard therapy)	30% reduction in risk of kidney failure or CV death (*p* < 0.001). HR for kidney failure progression: 0.70 (CI, 0.59–0.82)	The trial ended early due to apparent efficacy. Acute eGFR dips are common but not predictive of worse long-term renal outcomes
DAPA-CKD (NCT03036150) [44]	4304 adults with or without diabetes, CKD (eGFR 25–75 mL/min/1.73 m^2^, UACR 200–5000 mg/g)	Dapagliflozin vs. placebo (in addition to *standard therapy)*	36% reduction in the primary composite of ≥50% eGFR decline, ESKD, or kidney/CV death (*p* < 0.001). Benefitting diabetic & non-diabetic CKD	Lower rates of acute kidney injury, reinforcing a favorable safety profile
EMPA-KIDNEY (NCT03594110) [51]	6609 individuals with CKD (eGFR 20–45 or 45–90 mL/min/1.73 m^2^ with albuminuria ≥ 200 mg/g UACR)	Empagliflozin vs. placebo (in addition to *standard therapy)*	28% relative risk reduction in kidney disease progression or CV death (HR 0.72, 95% CI 0.64–0.82, *p* < 0.001)	Reinforced renal protection of SGLT2 inhibitors in diabetic and non-diabetic CKD, including those with mild albuminuria
VERTIS CV (NCT01986881) [52]	8246 adults with T2D and ASCVD (eGFR generally > 30 mL/min/1.73 m^2^, varying CKD stages)	Ertugliflozin vs. placebo (in addition to *standard therapy)*	Exploratory composite of sustained 40% eGFR decline, dialysis/transplant, or renal death significantly lower (HR, 0.66; CI, 0.50–0.88). UACR reduced by ~16–20%	Confirmed renal benefits of ertugliflozin in T2D with CV comorbidities, though primarily a CV safety trial

Abbreviations: ASCVD—Atherosclerotic Cardiovascular Disease; CI—Confidence Interval; CKD—Chronic Kidney Disease; CV—Cardiovascular; eGFR—Estimated Glomerular Filtration Rate (mL/min/1.73 m²); ESKD—End-Stage Kidney Disease; HR—Hazard Ratio; SGLT2—Sodium-Glucose Cotransporter-2; T2D—Type 2 Diabetes; UACR—Urine Albumin-to-Creatinine Ratio (mg/g).

**Table 3 biomolecules-15-00809-t003:** Summary of GLP-1 RA trials.

Trial	Patients	Intervention	Key Outcome	Notable Point
LEADER (NCT01179048) [62]	9340 individuals with T2D at high risk for cardiovascular events	Liraglutide (up to 1.8 mg daily) vs. placebo	13% reduction in MACE (HR 0.87), Significant reductions in cardiovascular death (HR 0.78), and all-cause mortality (HR 0.85)	Liraglutide not only reduced cardiovascular events but also decreased all-cause mortality in high-risk T2D patients.
FLOW (NCT03819153) [63]	3533 individuals with T2D and CKD (eGFR 25–75 mL/min/1.73 m^2^ and UACR 100–5000 mg/g)	Semaglutide 1.0 mg weekly vs. placebo	24% reduction in MAKE, Slowed annual eGFR decline by 1.16 mL/min/1.73 m^2^, Significant reductions in MACE (HR 0.82) & all-cause mortality (HR 0.80)	Semaglutide demonstrated significant benefits in reducing both kidney and cardiovascular events in patients with T2D and CKD.
SUSTAIN 6 (NCT01720446) [64]	3297 individuals with T2D at high risk for cardiovascular events	Semaglutide (0.5 mg or 1.0 mg once weekly) vs. placebo	26% reduction in MACE (HR 0.74); Significant reduction in non-fatal stroke (HR 0.61)	Semaglutide significantly reduced the risk of cardiovascular events in high-risk T2D patients.

Abbreviations: ASCVD—Atherosclerotic Cardiovascular Disease; CI—Confidence Interval; CKD—Chronic Kidney Disease; CV—Cardiovascular; eGFR—Estimated Glomerular Filtration Rate (mL/min/1.73 m^2^); ESKD—End-Stage Kidney Disease; HR—Hazard Ratio; MACE—Major Adverse Cardiovascular Event; MAKE—Major Adverse Kidney Event; SGLT2—Sodium-Glucose Cotransporter-2; T2D—Type 2 Diabetes; UACR—Urine Albumin-to-Creatinine Ratio (mg/g).

**Table 4 biomolecules-15-00809-t004:** Correlations between classifier scores for HF2, CAD160 and CKD273 and heart failure, coronary artery disease and chronic kidney disease events in a retrospective in silico analysis [88].

Classifier	Comparison	Unadjusted HR (*p*-Value)	Adjusted HR (*p*-Value)
HF2	Per 1-SD increment	2.59 (*p* < 2 × 10^−16^)	1.64 (*p* = 1.72 × 10^−18^)
HF2	Q5 vs. Q1	16.20 (*p* = 3.15 × 10^−39^)	3.84 (*p* = 5.64 × 10^−9^)
CAD160	Per 1-SD increment	1.72 (*p* < 2 × 10^−16^)	1.33 (*p* = 5.55 × 10^−7^)
CAD160	Q5 vs. Q1	4.73 (*p* = 4.93 × 10^−18^)	2.82 (*p* = 3.32 × 10^−8^)
CKD273	Per 1-SD increment	4.19 (*p* < 2 × 10^−16^)	3.18 (*p* = 1.03 × 10^−21^)
CKD273	Q5 vs. Q1	35.47 (*p* = 1.61 × 10^−16^)	19.59 (*p* = 7.32 × 10^−11^)

Abbreviations: CAD160—Coronary artery disease classifier with 160 peptides; CKD273—Chronic kidney disease classifier with 273 peptides; HF2—Heart Failure classifier version 2; HR—Hazard Ratio; Q1—First Quintile; Q5—Fifth Quintile; SD—Standard Deviation.

**Table 5 biomolecules-15-00809-t005:** Summary of biomarker studies in the systematic search.

Study	Patients	Intervention	Notable Point
Widiasta 2021 [91]	88 children with SRNS, 31 FSGS, 8 MPGN, 1 MesGN, 13 MCD, 62.5% male, age range 1–18 years.	CYC therapy, exclusion of calcineurin inhibitors.	TGF-β is crucial in SRNS treatment, high baseline TGF-β levels predict poor CYC response.
Teisseyre 2021 [92]	68 patients with primary membranous nephropathy	RTX therapy, administered as two 1 g infusions two weeks apart, was evaluated for its efficacy in achieving clinical remission in pMN patients.	Serum RTX levels predict clinical remission at months 6 and 12; undetectable RTX levels at month 3 indicate higher treatment failure risk.
Shiratori-Aso 2022 [90]	62 patients diagnosed with TIN	Corticosteroids and/or immunosuppressants for autoimmune TI	Elevated serum sIL2R levels may predict therapeutic response in autoimmune TIN
Kapsia 2022 [95]	100 patients with biopsy-proven PLN, mean age 31 ± 13 years, 80% female, all meeting 2019 classification criteria for SLE.	Drug therapy comparison between CYC and MPA as induction treatments, assessment of effects on kidney response, flares, and long-term outcomes in PLN patients.	Baseline proteinuria <1.5 g/day predicts time to complete response, 12-month proteinuria > 0.8 g/day correlates with flare occurrence, and interstitial fibrosis/tubular atrophy > 25% predicts long-term outcomes.
Jamee 2022 [93]	50 pediatric patients with NS, a mean disease follow-up duration of 3.6 years	Corticosteroid therapy as the primary treatment for pediatric NS, evaluating its impact on NLR/PLR.	No significant correlation between NLR/PLR ratios and steroid response.
Jaimes Campos 2023 [88]	5585 datasets were extracted, participants with urine samples at the baseline visit. Demographic covariables assessed included body mass index, age, sex, blood pressure, and eGFR. Median follow-up period: 3.74 ± 3.36 years.	Prediction of most beneficial interventions in CKD, HF, and CAD, the following interventions were investigated: MRA, SGLT2i, DPP4i, ARB, GLP1RA, olive oil, and exercise.	Significant effects of treatments on in silico urinary peptides observed. Findings support personalized strategies for cardiovascular and kidney disease management. Prospective clinical trial validation is needed for clinical utility assessment.
Jaimes Campos 2024 [86]	Discovery cohort (DCREN): 199 adults treated with RAS inhibitors. PRIORITY cohort: 1078 participants with T2D selected for analysis (not receiving spironolactone). DIRECTProtect 2 cohort: 1905 individuals with T2D, 365 treated with candesartan.	RAAS blocking agents studied in diabetic patients to prevent DKD progression.	DKDp189 model predicts nonresponse to RASi treatment in diabetic patients. Urinary peptides may serve as biomarkers for DKD progression. Study highlights the variability in eGFR classification methods.
Capelli 2023 [96]	Patients aged > 18 years with T2D and CKD stages G2 and G3.	SGLT2i in patients with T2D and proteinuric CKD, evaluating effects on proteinuria reduction and baseline predictors of response.	SGLT2i therapy reduced proteinuria by >30% in most patients. Baseline proteinuria, eGFR & BMI are key predictors of treatment response.
Heerspink 2021 [97]	3668 adults with T2D and CKD (eGFR 25–75 mL/min/1.73 m², UACR 300–5000 mg/g), 98.5% on ACE/ARB therapy.	Atrasentan 0.75 mg/day added to background RAS blockade, employing a six-week “response enrichment” phase based on UACR reduction.	Early UACR response was not a causal predictor of atrasentan’s long-term kidney protection.

Abbreviations: ACE—Angiotensin-Converting Enzyme; ARB—Angiotensin II Receptor Blocker; BMI—Body Mass Index; CAD—Coronary Artery Disease; CKD—Chronic Kidney Disease; CR—Complete Remission; CYC—Cyclophosphamide; DKD—Diabetic Kidney Disease; DPP4i—Dipeptidyl Peptidase-4 Inhibitor; eGFR—Estimated Glomerular Filtration Rate; FSGS—Focal Segmental Glomerulosclerosis; GLP1RA—Glucagon-Like Peptide-1 Receptor Agonist; HF—Heart Failure; MCD—Minimal Change Disease; MPA—Microscopic Polyangiitis; MRA—Mineralocorticoid Receptor Antagonist; MesGN—Mesangial Glomerulonephritis; MPGN—Membranoproliferative Glomerulonephritis; NLR—Neutrophil-to-Lymphocyte Ratio; NS—Nephrotic Syndrome; PLR—Platelet-to-Lymphocyte Ratio; PLN—Proliferative Lupus Nephritis; RAS—Renin-Angiotensin System; RASi—Renin-Angiotensin System Inhibitor; RTX—Rituximab; SGLT2i—Sodium-Glucose Cotransporter-2 Inhibitor; SLE—Systemic Lupus Erythematosus; SRNS—Steroid-Resistant Nephrotic Syndrome; T2D—Type 2 Diabetes; TGF-β—Transforming Growth Factor Beta; TIN—Tubulointerstitial Nephritis; UACR—Urine Albumin-to-Creatinine Ratio.

**Table 6 biomolecules-15-00809-t006:** Summary of the approximate costs in Germany for the primary interventions mentioned.

Therapy	Monthly Cost (€)	Annual Cost (€)	Notes
ACE Inhibitor (Ramipril 5 mg daily)	4.20	50.40	Very inexpensive generic; ~€0.14 per unit
ARB (Losartan 50–100 mg daily)	6.90	82.80	Slightly more expensive, generic; ~€0.23 per unit
Ns-MRA (Finerenone 10–20 mg daily)	61.20	734.30	Brand drug; ~€2.04 per unit
SGLT2 inhibitor (Empagliflozin 10–25 mg daily)	57.30	687.60	Brand drug; ~€1.91 per unit
Long-acting GLP-1 RA (Semaglutide 1 mg once weekly)	312.75	3755.96	Brand drug; ~€72.23 per injection solution
Sparsentan 400 mg (for IgA nephropathy, once daily)	4935.94	59,231.28	Brand drug; ~€164.53 per unit
Dialysis [108]	~3916	~47,000	Peritoneal & Hemodialysis cost approximately the same

Prices are in Euros (€) and based on the website www.shop-apotheke.com (accessed in 12 March 2025). Abbreviations: ACE Inhibitor—Angiotensin-Converting Enzyme Inhibitor; ARB—Angiotensin II Receptor Blocker; GLP-1 RA—Glucagon-Like Peptide-1 Receptor Agonist; ns-MRA—Non-steroidal Mineralocorticoid Receptor Antagonist; SGLT2—Sodium-Glucose Cotransporter-2.

## Data Availability

No new data were created or analyzed in this study. Data sharing does not apply to this article.

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
