# Peer review of "The Future of Chronic Kidney Disease Treatment: Combination Therapy (Polypill) or Biomarker-Guided Personalized Intervention?"

_biomolecules, 2025, doi:10.3390/biom15060809_

Round 1
Reviewer 1 Report
Comments and Suggestions for Authors
In this review article (Biglari et al), the authors systematically assess advances in pharmacotherapy for chronic kidney disease (CKD) between 2020 and 2025, emphasizing combination therapies and biomarker-guided interventions.
The authors discussed various therapeutic strategies, including RAAS blockers, sodium–glucose cotransporter 2 (SGLT2) inhibitors, glucagon-like peptide-1 (GLP-1) receptor agonists, and non-steroidal mineralocorticoid receptor antagonists (ns-MRAs).
However, the authors omit the discussion of dipeptidyl peptidase-4 (DPP-4) inhibitors in the paragraph "Therapeutic Landscape of CKD Management" without offering a clear justification. It is known in literature that in patients with advanced CKD, treatment with DPP-4 inhibitors in combination with sodium–glucose cotransporter 2 (SGLT-2) inhibitors provide greater renal benefits.
Therefore, it could be better if the author included this therapy—see reference PMID: 38545319—or providing a rationale for its exclusion.
Overall, this review is well composed and based on a significant background.
Author Response
We thank the reviewer for raising this point. In framing the Therapeutic Landscape section we applied two pre-specified filters that unintentionally excluded DPP-4 inhibitors:
- Outcome-driven inclusion criterion: We concentrated on drug classes for which randomised trials published 2020-2025 demonstrated a statistically significant reduction in CKD progression or kidney failure. This criterion is stated in the Methods and was introduced to keep the review focused on therapies with proven disease-modifying efficacy rather than on glucose-lowering per se.
- Citation impact threshold: Articles had to reach ≥10 citations per year to be fully appraised (Methods). Within the 2020-2025 window, no DPP-4 trial that met renal-outcome and citation thresholds was identified.
Although the 2024 meta-analysis cited by the reviewer (Mahzari et al., PMID 38545319) confirms that DPP-4 inhibitors improve HbA1c and are safe in advanced CKD, the pooled trials were not powered for kidney end-points; the meta-analysis itself reports no significant effect on albuminuria, eGFR decline, or hard renal outcomes. Consequently, recent KDIGO guidelines (2024 CKD general guideline and 2022 Diabetes-in-CKD guideline) list DPP-4 inhibitors as glycaemic agents while reserving “disease-modifying” status for RAAS blockade, SGLT-2 inhibitors, GLP-1 receptor agonists, and ns-MRAs.
For these reasons we believe a detailed discussion of DPP-4 inhibitors would broaden the article beyond its declared aim, to evaluate therapies with direct reno-protective evidence and their use in combination or biomarker-guided strategies.
Reviewer 2 Report
Comments and Suggestions for Authors
The manuscript "The Future of Chronic Kidney Disease Treatment: Combination Therapy (Polypill) or Biomarker-Guided Personalized Intervention?" presents an intriguing effort by the authors to integrate existing knowledge with current clinical research regarding strategies to prevent and/or postpone end-stage renal disease in patients with CKD. Although the authors noted that they restricted their research to papers published within the last five years on the Web of Science with at least 10 citations, this manuscript provides remarkable insights into the advancement of therapies focused on decelerating the progression of chronic kidney disease. The authors also examined studies involving combined pharmacological treatments and biomarker-guided interventions. The authors provided their insights on cutting-edge strategies for personalized treatment, highlighting both the clear benefits and the drawbacks that accompany them. An outstanding review.
Author Response
We are very grateful for the reviewer’s encouraging assessment of our work. We appreciate that the reviewer found the synthesis of current combination strategies and biomarker-guided approaches useful and that the restricted‐timeframe, citation-based search strategy was judged appropriate.
Reviewer 3 Report
Comments and Suggestions for Authors
This is a complete and very well written review. This review addresses the topic of biomarker-guided personalized intervention in chronic kidney disease. The topic is very original, from a precision medicine perspective. It is a contemporary topic. As far as I know there is no similar published material. Of course, Figure 1 conveys a known scheme in studies related to chronic kidney disease, but it is necessary for general readers. The methodology for the selection of original studies is carefully detailed. From the beginning, the authors collect ideas and results and constructed very well-structured conclusions/recommendations. The figures are very clear. The references are very complete and interesting selection of original studies.
Author Response
We kindly thank the reviewer for the very positive appraisal of our manuscript and for recognising the novelty of a precision-medicine-centred overview in chronic kidney disease. We appreciate the reviewer’s endorsement of the manuscript’s structure, figures, and reference selection.
Reviewer 4 Report
Comments and Suggestions for Authors
This review comprehensively evaluates the efficacy of a combination therapy (polypill) versus a biomarker-guided personalized intervention for CKD patients.
Medications for CKD include ACE inhibitors and ARBs for blood pressure control, statins for cholesterol, a diuretic, and possibly something like SGLT2 inhibitors, which are newer and show promise for this disease. The idea is that by combining these treatments, treatment will be simplified, adherence will be improved, and patients will receive all the necessary medications.
By contrast, biomarker-guided personalized interventions use specific biomarkers to tailor treatment for each patient. CKD biomarkers include albuminuria (urine albumin-to-creatinine ratio), serum creatinine, eGFR, and perhaps novel biomarkers such as NGAL (neutrophil gelatinase-associated lipocalin), KIM-1 (Kidney Injury Molecule-1). These biomarkers would be used to adjust treatments based on the personalized approach. Targeting specific pathways in the patient's disease could improve efficacy and reduce side effects by avoiding unnecessary medications.
Each strategy has its pros and cons. Polypills offer simplicity, better adherence, lower costs, and broad coverage across multiple pathways. Some patients may overtreat, causing side effects from medications they don't need. Statins are not appropriate for all CKD patients, and ACE inhibitors may not be tolerated by all. Furthermore, fixed doses might not be optimal for everyone. It is important for the authors to discuss the advantages (precision, dynamic adjustments, potential for better outcomes versus) and disadvantages (cost, evidence gaps, and infrastructure requirement) of biomarker-guided personalized intervention.
The author may like to include the following relevant information.
- Polypill Trials: UK HARP-III demonstrated feasibility but focused on cardiovascular risk, not CKD-specific outcomes.
- Biomarker-Guided Trials: PRIORITY trial reduced ESRD risk in diabetic CKD using proteomic-guided interventions.
- Contextual Considerations - Advanced or diabetic CKD may benefit more from personalization, while early-stage patients gain from polypill simplicity.
- Guidelines: KDIGO recommends ACE/ARB and SGLT2 inhibitors broadly, but emerging data support biomarker stratification.
- Future direction - may need a section to discuss the hybrid system. Base polypill for risk factor control, augmented by biomarker-guided add-ons (e.g., GLP-1 agonists for diabetic CKD with inflammation).
Author Response
We sincerely thank the reviewer for the thoughtful, detailed appraisal of our manuscript and for the specific suggestions that have helped us clarify and strengthen the work. Below we reproduce each comment followed by our point-by-point response and a summary of the changes made to the revised version (tracked in the manuscript).
In response to reviewer’s suggestions, we have made the following revisions:
- Polypill Feasibility in CKD:
We have clarified that while polypill strategies have been shown to be feasible in the context of cardiovascular disease their use in CKD has historically been limited by adding the following paragraph to the beginning of the Advantages & Disadvantages of Combination Therapy section (page 15):
“Large trials have demonstrated a clinical value of fixed-dose combination therapy in the context of cardiovascular disease[81–83], yet their use in CKD has historically been limited by the lack of multiple effective disease-modifying therapies. However, recent advances such as the emergence of SGLT2 inhibitors as foundational therapy alongside RAAS blockade have shifted this landscape. These developments now make the implementation of polypill strategies in CKD a feasible and promising avenue”.
- CKD273 and Predictive Biomarkers:
We expanded the section on biomarkers to include additional details on the PRIORITY study, which demonstrated that the proteomic classifier CKD273 can identify patients who may benefit from earlier intervention. Following text is now included in the Discussion (page 22):
“In the PRIORITY trial, a CKD273 score of 0.154 identified 12.1% of normo-albuminuric T2D patients with a 2.48-fold (CI: 1.51–4.08; P<0.001) increased risk to develop albuminuria, after adjusting for baseline clinical factors [97,98]. This demonstrates that the proteomic classifier CKD273 enables identifying patients who may benefit from earlier intervention”
- Tailored vs. Simplified Treatment Strategies:
We added further discussion distinguishing the utility of treatment approaches by CKD stage. We have included the following sentences in the hybrid approach section of the Discussion (page 23):
“For example, in early-stage CKD a streamlined yet unguided fixed-dose polypill built around an SGLT2 inhibitor plus an ARB may offer practical benefits by delivering broad, guideline conforming protection (upon validation in a properly powered clinical trial). Conversely, in advanced CKD where patients are at massively increased risk of developing ESKD and/or experiencing cardiovascular events, personalized approaches appear to be the superior choice, as these are expected to lead to better out-comes and spare patients’ unnecessary polypharmacy or complications.”
- Guideline Perspective on Biomarkers:
We included the following clarification in the revised text, as the last paragraph of the Future Directions and Clinical Implications section (page 22):
“While there is growing interest in leveraging additional biomarkers (beyond eGFR and albuminuria) to enable more precise risk stratification and individualized therapy, current KDIGO guidelines do not yet formally endorse novel biomarkers for routine clinical decision-making. Nonetheless, the guidelines acknowledge the potential for biomarker-guided interventions as emerging evidence solidifies their clinical utility.”
- Hybrid Approaches:
Responding also to comment 3, we elaborated on the potential for a hybrid approach that combines the strengths of both strategies, initial broad implementation of polypills to improve adherence and coverage, with the option for stepwise personalization in patients identified (via biomarkers) as requiring tailored interventions (please see above for the paragraph added to the manuscript).
Round 2
Reviewer 1 Report
Comments and Suggestions for Authors
I thank the authors for their response, the paper is accept in present form.